# Quantifying the topographical structure of rocky and coral seabeds

**Damien Sous**[1,2]*, **Samuel Meulé**[3], **Solène Dealbera**[1], **Héloïse Michaud**[4], **Ghislain Gassier**[3], **Marc Pezerat**[5], **Frédéric Bouchette**[6]

1 Université de Pau et des Pays de l'Adour, E2S-UPPA, SIAME, Anglet, France, 2 Université de Toulon, Aix Marseille Univ, CNRS, IRD, MIO, Marseille, France, 3 Aix Marseille University, CNRS, IRD, INRAE, Coll France, CEREGE, Aix-en-Provence, France, 4 Shom, Antenne de Toulouse, Toulouse, France, 5 Shom, Antenne de Brest, Brest, France, 6 GEOSCIENCES-Montpellier, Univ Montpellier, CNRS, Montpellier, France

☋ These authors contributed equally to this work.
* damien.sous@univ-pau.fr

**Data Availability Statement:** The data is avalaible online at https://doi.org/10.17882/98753.

**Funding:** The author(s) received no specific funding for this work.

## Abstract

Describing the structural complexity of seabeds is of primary importance for a number of geomorphological, hydrodynamical and ecological issues. Aiming to bring a decisive insight on the long-term development of a unified view, the present study reports on a comparative multi-site analysis of high resolution topography surveys in rough nearshore environments. The nine study sites have been selected to cover a wide variety of topographical features, including rocky and coral seabeds. The topography data has been processed to separate roughness and bathymetry-related terrain features, allowing to perform a comprehensive spectral and statistical analysis of each site. A series of roughness metrics have been tested to identify the most relevant estimators of the bottom roughness at each site. The spectral analysis highlights the systematic presence of a self-affine range of variable extension and spectral slope. The standard deviation of the seabed elevation varies from 0.04 to 0.77 m. The statistical and multi-scale analysis performed on the whole set of roughness metrics allows to identify connection between metrics and therefore to propose a reduced set of relevant roughness estimators. A more general emphasis is placed on the need to properly define a unified framework when reconstructing roughness statistics and bathymetry from fine seabed topographical data.

## Introduction

Rocky and coral seabeds display a striking variety of geometrical structures, evolving under the continuous action of geomorphological processes (erosion and deposition) and biogeochemical activity which are both particularly intense in nearshore areas. The description of the geometrical structure of rough seabeds is of primary importance for three main topics. First, from an ecological view, the seabed structure has been early recognized as a key factor for ecosystem richness and health [1, 2]. This is particularly critical for coral reef ecosystems which display, when fully healthy, both spectacular geometries and unrivalled biodiversity [3, 4]. A positive correlation is now established between the architectural complexity, understood as the level of

**Competing interests:** The authors have declared that no competing interests exist.

topographical irregularity and heterogeneity of a given site [5, 6], and the coral health [3] and resilience [7], and the related ecosystem biodiversity [8]. The combined pressures of human action and global change are expected to deleteriously affect the architectural complexity and, by extension, the whole ecosystem dynamics [9, 10]. Second, from a hydrodynamical aspect, the bottom roughness is a key driver of frictional dissipation which affects the momentum-related parameters, such as currents and water levels [11, 12], and the propagation of the wave field which in turn controls the wave energy transfer and wave-induced impacts at the shore [13–15]. A few studies have performed combined analysis of wave propagation and fine seabed topographical measurements in order to draw relationship between the wave attenuation and the seabed roughness structure [16–20]. The common finding was that the dominant scaling of roughness structure is the standard deviation of the seabed elevation. However recent observations may lead to assume that other seabed parameters may affect bottom friction [20]. In this context, a core issue is the development of quantitative characterization of the seabed structure, a prelude to the implementation of roughness-metrics-based rather than empirical parameterizations of wave frictional dissipation over rough coastlines worldwide. Third, the topographical structure is an overall dynamic marker of the geomorphological evolution in response to environmental forcings and biogeochemical processes [21–24]. This is particularly significant in coral reef environments, which display a strong reactivity to environmental conditions [6, 25], but also marked in rocky areas at a longer time scale.

Anticipating the future of rocky and coral coastlines, and their ecosystems, in the present context of growing vulnerability requires the development of accurate and usable tools for quantifying the seabed structure, in the long-term prospect of a global survey [26, 27]. Acoustic measurements have been widely performed to characterize seabed roughness structure and its feedback on acoustic bottom sensing techniques [28–33], sedimentary relief dynamics [34], benthic habitat [35–37], vegetation [38] and, at much larger scale, geological and tectonical issues [39, 40]. Laser line measurement allows to reach much higher resolution, i.e. 1–10 mm, but remains very local [41, 42]. The recent development of fine 3D survey techniques, such as photogrammetry, laser scanning or high-resolution acoustic sensing possibly mounted on autonomous vehicle [24, 43–46], have spectacularly increased the accuracy ($\mathcal{O}(1 - 10cm)$) and spatial coverage ($\mathcal{O}(10 - 100m)$) of high-resolution seabed topography. However, this new influx of 3D dataset raises several questions. First, most of the proposed technologies remain highly costly in terms of instrumentation and technical expertise, which excludes their use by local communities. Moreover, the volume of generated data can become a limiting factor when trying to survey the large-scale range of terrain features, which can be challenging to document at high-resolution despite their potential weight in topographical metrics computation. In particular, the scale range discrimination of roughness patterns is of primary importance for circulation and wave numerical models, which need to differentiate between bathymetry and roughness. This choice, often arbitrary and/or based on purely numerical constraints, is further debated in the Discussion section of the present paper. Ecological research literature on architectural complexity displays a wide range of methods and scale, from quantitative direct sub-centimetric measurements over a few meter domain [6] or lidar scanning over kilometric area with a metric resolution [47] to qualitative visual classification of seabed types over a few dozen of meters [48]. Hence, the large-scale range of a given study can be the small-scale one for another. More generally, there is a lack of unification in the measurement methods and the definition of the topographical structure itself, which, as raised by [6], may vary depending on the type of recovered data, the studied scales, the targeted issue or the scientific community involved, and the metrics used to quantitatively differentiate between sites, or simply their names. There is now a crucial need for representative metrics applicable to real seabeds, established on a wide range of spatial scales. Note also that, as shown by the

disproportionate available literature, most efforts in quantifying the architectural complexity have been dedicated to coral reef systems. While not as spectacular as coral reefs in terms of biodiversity and ecosystem dynamics, we are convinced that rocky seabeds should deserve more attention and, more generally, that the establishment of roughness geometry metrics should benefit for more unified and comprehensive view integrating each type of benthic substrate. This study aims at filling these identified gaps and contributes to advancing the understanding of seabed structure.

The objective of the present study is therefore to identify, over a selected sample of field sites, the relevant metrics for quantifying the geometrical structure of seabeds, with a particular focus on nearshore shallow regions. The overall motivation of the investigation is to answer the need to find representative estimators able to quantitatively discriminate between seabed types. The guidelines are (i) to select metrics which are practically calculable for real seabeds and measurable with standard topographical surveys and (ii) to document a large range of spatial scales (typically three spectral decades) to be able to connect a continuum of terrain pattern scales [49] and to move toward a unified view between the approaches used in hydrodynamical, geomorphological and ecological fields. A series of nine sites have therefore been selected, showing a wide variety of geological context (rocks and coral) and roughness scales. A particular effort has been paid on establishing an unified selection of metrics, often used with different names in different scientific fields, as developing practical and scalable metrics that can be widely applicable is a priority.The first part of the paper is dedicated to the presentation of the field sites together with a description of topographical data acquisition and processing. The second part presents the results, including a comprehensive inter-sites comparison, while the last part provides discussion and recommendations for further developments and applications.

## Materials and methods

The present section describes the nine sites selected to cover a wide variety of roughness structure, the methods used for producing high-resolution topographic surveys on each site and for processing the topographical data. The Matlab ™code used to process the data is provided in S1 Appendix.

### Field sites

**Socoa.**   The *Socoa* site is located on the Atlantic coast, in the Basque Country near the French-Spanish border (Fig 1A and 1B). The studied intertidal rocky platform is located west of the Saint Jean de Luz bay. The platform is approximately 100 m long. The shore platform and the associated cliff display the so-called *Flysch marno-calcaire de Socoa*, corresponding to a marl and limestone Flysch formation [50, 51].

The roughness geometry displays a peculiar structure, characterized by along-shore oriented ridges with a typical height ranging from 0.2 to 0.8 m in the studied area (Fig 1C and 1D). The ridges show a marked cross-shore-asymmetry with 30 to 60˚ sloping upstream faces and much steeper downstream faces.

**Ars en Ré.**   The *Ars en Ré* site covers an intertidal rocky platform located on the Atlantic coast, along the north-west coast of the Ré island, France(Fig 2A and 2B). The studied site is the lower part of the foreshore, which is a large rocky flat, made of marl and argillaceous limestone, while the upper part of the foreshore is covered by a narrow sandy strip. In the studied area, the platform displays a 700 m-long subhorizontal (slope about 0.001) nearly along-shore uniform structure. Sparse pebbles and small boulders, with typical sized ranging from 0.1 to

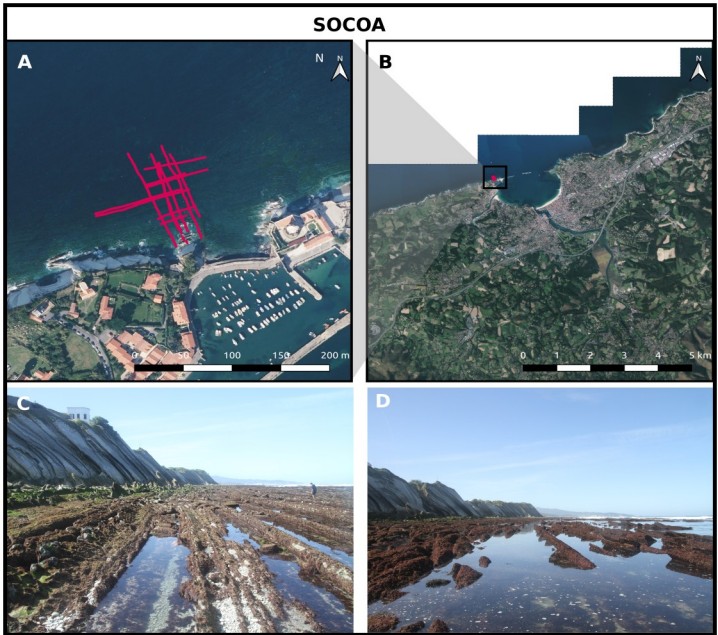

**Fig 1. The Socoa site.** A and B: aerial picture (BD ORTHO®, IGN©) of the Socoa site. Red tracks depict the GNSS surveys. C and D: Illustrations of the flysch reef platform at Socoa.

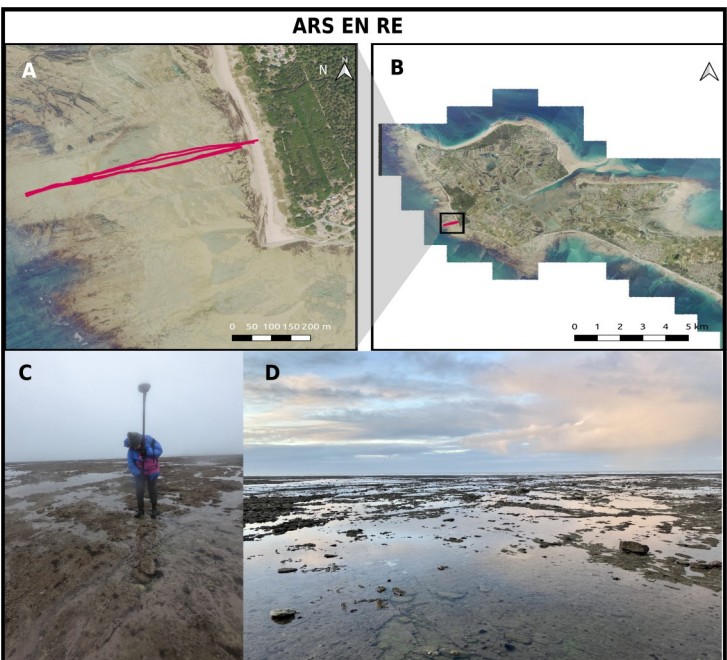

**Fig 2. The Ars en Ré site.** A and B: aerial pictures (BD ORTHO®, IGN©) of the Ré Island showing the surveys transects in red. B: Lidar-extracted digital elevation model. C and D: Illustrations of the rocky shore at Ars en Ré.

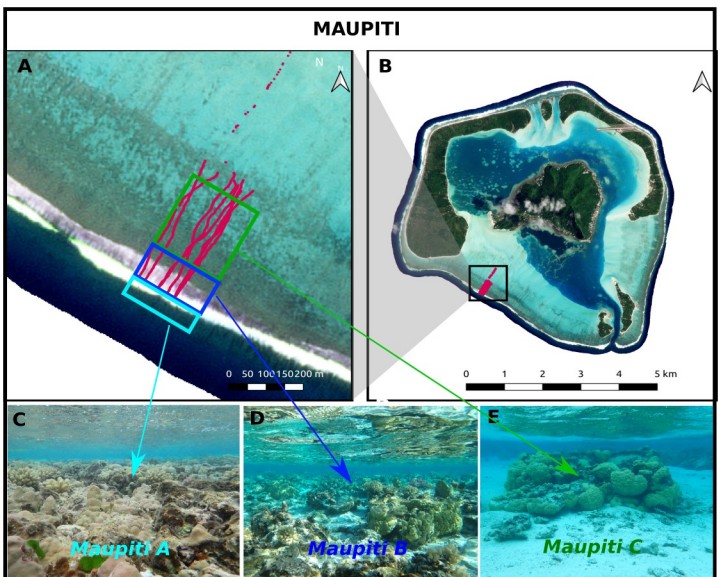

**Fig 3. The Maupiti site.** A and B: Satellite imagery © AIRBUS DS (2020) of the Maupiti barrier reef. Red tracks depict the GNSS surveys. C, D and E: Underwater pictures for sites Maupiti A, Maupiti B and Maupiti C, respectively.

0.5 m, are spread along the platform which displays a rather uniform roughness topography marked by sparse vertical steps (typical heights between 0.1 to 0.3 m).

**Maupiti.** The *Maupiti* site is located at the south-west barrier reef of Maupiti Island (Fig 3A and 3B). Maupiti is a diamond-shaped high volcanic island, located in the western part of the Leeward Islands, Society Archipelago, French Polynesia. The selected site has been documented, both topographically and hydrodynamically, by a series of recent field studies [12, 20, 49]. The reef barrier displays a nearly uniform along-reef structure but a striking cross-shore partitioning in terms of reef colony structure (Fig 3C, 3D and 3E). Three distinct zones of the nearly horizontal back-reef, namely Maupiti A, B and C, are selected for the present study. Maupiti A is marked by a low-crested compact reef structure while moving into Maupiti B and C, the reef is increasingly open with higher, larger and more scattered reef pinnacles.

**Niau.** The island of *Niau* is part of the UNESCO Fakarava Biosphere Reserve. It is an atoll in the Tuamotu archipelago, French Polynesia. The volcano was formed in the Eocene [52] and shows signs of subsidence [53] and erosion. As a result, the land surrounding a totally enclosed lagoon is higher (around 7.5 m and up to 15 m in the northeast) than the current ocean level. The study area is located to the north of the village of Tupana, in the northeastern part of the lagoon (Fig 4A and 4B). It is a fringing reef extending from the reef crest to the beach, on a 35 m long horizontal reef flat. The reef crest is a typical algal ridge intersected by small "hoas" running towards the front reef where live coral is present and varied. From the upper reef, covered with patches of algea, to the sandy back reef, the very shallow water zone consists of a dense, spatially uniform pavement of dead coral (Fig 4C and 4D) associated with very few small ponds.

**Parlementia.** The *Parlementia* sites are located on the french Atlantic coast, in the Basque Country about 8km northeastward from the Socoa site (Fig 5A and 5B). The studied intertidal rocky platform is located east of the Guethary harbor. The platform display the so-called *flint-Flysch of Guethary*, characterized by a layering of calcarenites, calcilutites and marl [54, 55]. Two sites, namely Parlementia A and B, have been selected on the platform. Parlementia A

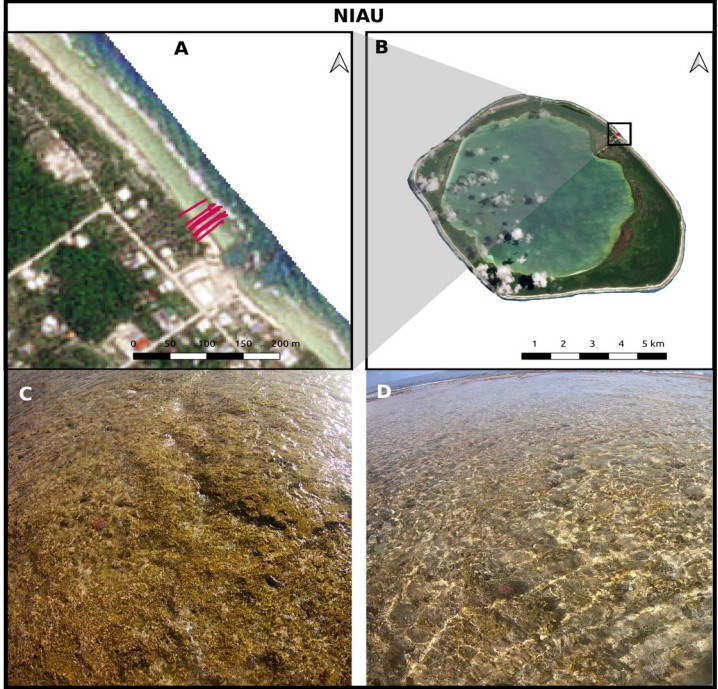

**Fig 4. The Niau site.** Red tracks depict the GNSS surveys. A: Satellite imagery Pléiades © CNES (2019), AIRBUS DS Distribution. B: Satellite imagery Copernicus Sentinel 2021, processed by ESA CC BY-SA 3.0 IGO. C and D: Illustrations of the flat back reef.

corresponds to the wide platform detached from the shore by a shallow sandy lagoon. The seabed is here made of strongly rounded flysch rocks mixed with rolled boulders and pebbles, with a typical size ranging from 0.05 to 0.3m in the studied area (Fig 5C and 5D). Parlementia B corresponds to the fringing reef connected to the beach. It displays a much higher, rugged and sharp morphology, with slopping and detached flysch plates up to 2 m high.

**Banneg.** The *Banneg* site is located on the western coast of the Banneg Island, Molène Archipelago, France. This small inhabited island is oriented north-south, 0.8 km long and 0.15 to 0.35km wide (Fig 6A). The western coast (Fig 6C) displays steep cliffs ($\tan \beta > 0.5$) that include a series of high headlands (16 m to 20 m above mean sea level, MSL), and lower cliffs (12 m to 13 m above MSL) with more gentle slopes ($0.15 < \tan \beta < 0.4$) in embayments. These cliffs present an orthogonal tabular structure resulting from the horizontal bedding and nearly vertical joint system affecting the granite bedrock [56]. Under the effect of extreme forcing, coarse deposits made of rock slabs torn from the substratum, the so-called *cyclopean blocks*, are dispersed along the upper shoreline [57].

### Topographic surveys

The topographic data analysed here are based on two types of measurement strategies, see Table 1 for a summary.

First, most of the processed data relies on a series of on-foot GNSS (Global Navigation Satellite System) surveys. These measurements, used for Socoa, Ars en Ré, Parlementia, Maupiti and Niau, are based on the method described in [49]. Other techniques may have been used [45, 58] but, while time and efforts consuming, the on-foot GNSS approach remains easily deployable on each site whatever the meteorological conditions are and allows to cover both

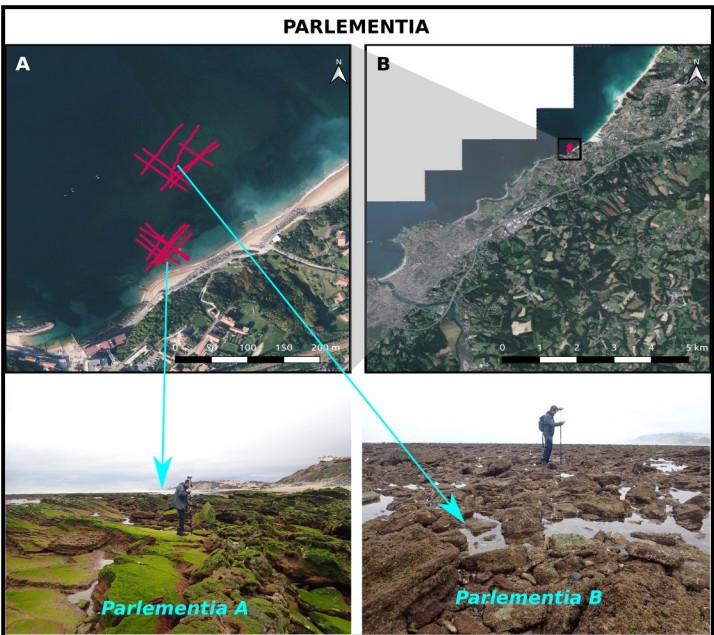

**Fig 5. The Parlementia sites.** A and B: aerial picture (BD ORTHO®, IGN©) of the Parlementia sites. Red tracks depict the GNSS surveys. C and D: illustration of the rocky sites Parlementia A and B, respectively.

emerged and submerged parts of the selected areas. The GNSS rovers are used in Real Time Kinematics, either communicating with a fixed base (TRIMBLE R8 and R8S) or through network RTK (Leica GS14). The acquisition modes were at a fixed acquisition period of 1 s. The typical displacement velocity of the operator ranged between 2 and 8 cm/s, leading to

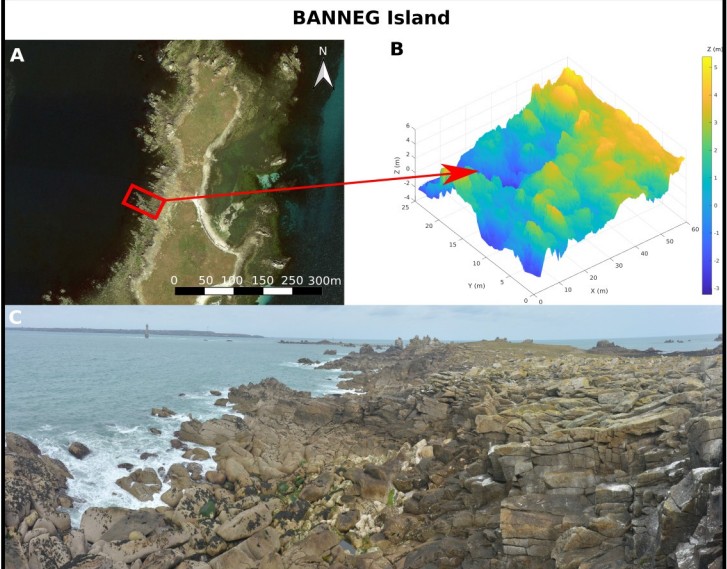

**Fig 6. The Banneg site.** A: aerial picture (BD ORTHO®, IGN©) of the Banneg Island showing the zone of interest (red rectangle). B: Lidar-extracted digital elevation model. C: Illustration of the Banneg west coast shore.

**Table 1. Topographic survey parameters.** CS and AS refer to cross-shore and along-shore transects, respectively.

| Site | Survey type | Number of surveys | Average resolution (m) |
|---|---|---|---|
| Socoa | On-foot GNSS | 5 CS + 6 AS | 0.037 |
| Ars en Ré | On-foot GNSS | 6 CS | 0.071 |
| Maupiti A | On-foot GNSS | 4 CS | 0.074 |
| Maupiti B | On-foot GNSS | 4 CS | 0.074 |
| Maupiti C | On-foot GNSS | 4 CS | 0.074 |
| Niau | On-foot GNSS | 6 CS | 0.041 |
| Parlementia A | On-foot GNSS | 4 CS + 4 AS | 0.038 |
| Parlementia B | On-foot GNSS | 3 CS + 4 AS | 0.043 |
| Banneg | Airborne Lidar | 250 CS + 600 AS | 0.12 |

horizontal resolution between 2 and 8cm (Table 1). The typical accuracy of these on-foot GNSS surveys is 15 and 3 cm for horizontal and vertical directions, respectively [49].

The second approach is based on a airborne LIDAR survey. It has been used to monitor the Banneg site, due to the difficulty of on-foot access in this harsh environment. LIDAR surveys exploited in this study were carried out within the framework of Litto3D® program. This national program is based on a partnership between Shom and the French National Geographic Institute (IGN) [59]. It aims to provide very high resolution coastal altimetric models of metropolitan and overseas French coasts [60]. Topo-bathymetric data were acquired from a BLOM CGR and CAE Aviation type aircraft equipped with an airborne lidar bathymetric LADS Mk III and topo-bathymetric RIEGL VQ-820-G in spring 2012 and 2013.

## Data processing

The general principle of the processing is, for a given site, to treat each transect individually and then average the individual results over the complete series of transects, in order to get more statistically-robust metrics. The first step of the data preparation is a reduction of the spatial extent to focus on the common areas of interest for each profile, discarding zones of low measurement resolution. Then, each profile is interpolated on a regular grid with a 0.05m horizontal step selected to discard any visible modification of the spectra in the studied ranges. Similar processing is applied on the Banneg lidar data, with a selection of a high resolution 60x25 m area and a linear interpolation on a regular grid. Each row of the gridded data is then considered as an individual transect.

The data processing consists first in a spectral analysis of the interpolated seabed signal. For the GNSS transect, a fourier transform is applied on each interpolated profile. In order to get an accurate characterization of the low wavenumber content, no windowing is applied and a low-wavenumber cut-off is fixed corresponding to one third of the total profile length. Statistical stability is increased by merging over 7 wavenumbers [61], resulting in 16 degrees of freedom and spectral resolution of 0.01. For each site, the recovered spectral densities of variance are averaged over the set of monitored transects, in order to get an overall view of the site. Similar processing is applied on the Banneg lidar data, based on the combined analysis of the cross-shore profiles extracted from the gridded data.

A statistical analysis is then performed to produce histograms and several topographical metrics on seabed elevation (see next section for details). The bathymetry is extracted from each transect, following the moving-window distribution method proposed by [12]. Specifically, the reference bathymetry is defined as the 10-th percentile of the reef elevation in each successive window. The moving-window length is here fixed at 10 m. This approach preserves

topographical wave-lengths larger than the window, which are deemed bathymetry terrain features, while smaller length-scales are considered as roughness terrain features which are extensively characterized in the present statistical analysis. This choice is further assessed in the Discussion section. The extracted bathymetry $Z_r$ is finally removed from the raw seabed elevation signal $z$ to produce $z_b = z - Z_r$, the corrected seabed elevation from which a series of topographical metrics are extracted. Fig 7 displays illustrative profiles of raw seabed, bathymetry and corrected seabed for Socoa and Maupiti C sites.

## Topographical metrics

Combining standard statistical moments and roughness estimators proposed in the literature [6, 21–23, 45, 49, 62–67], a series of metrics are computed for each transect and then transect-averaged to obtain the proposed values for each site. The only exception is the directionality index (Eq 11) which is computed from transect-averaged metrics.

**Spectral slope $\beta$ and saturation wave-number $K_s$.**   Many terrestrial and extra-terrestrial environments display a typical spectral shape, combining three idealized ranges: a saturation flat-spectrum range at low wave-numbers, a fractal-type power-law at medium wave-numbers (also called self-affine) and a smooth region at high wave numbers where the spectrum declines rapidly to zero [21, 22, 49, 62]:

$$S = \begin{cases} \approx cst & K < K_s \\ \propto K^{-\beta} & K_s < K < K_{s2} \end{cases} \tag{1}$$

where $S$ is the spectral density of variance of seabed elevation, $K$ is the wave-number, $\beta$ the spectral slope in the self-affine range and $K_s$ and $K_{s2}$ the upper and lower limits of the self-affine range. For a given total variance of seabed elevation, $K_s$ and $\beta$ will have a strong impact on the topographical structure, the former describing the limit size of the large-scale terrain features (large/small $K_s$ inhibiting/allowing the presence of large patterns) while the latter quantifies the spectral distribution of roughness length-scale (negatively large/small $\beta$ decreasing/increasing the small- to large scale features ratio).

**Mean elevation.**   The most basic estimator of the roughness height is the mean elevation, defined as the arithmetic mean of seabed elevation along the selected transect:

$$z_m = \frac{1}{n} \sum_{i=1}^{n} z_{b,i} \tag{2}$$

where $n$ is the number of data points along the transect.

**Standard deviation.**   The standard deviation of the bed elevation, $STD$, is defined as:

$$STD = \sqrt{\frac{1}{n} \sum_{i=1}^{n} (z_{b,i} - z_m)^2} \tag{3}$$

A significant connection may be expected between $z_m$ and $STD$, which both describe the roughness height from area integrals. Note that other roughness height metrics based on peak-to-trough measurements may be more subject to corruption by extreme elevation values [63].

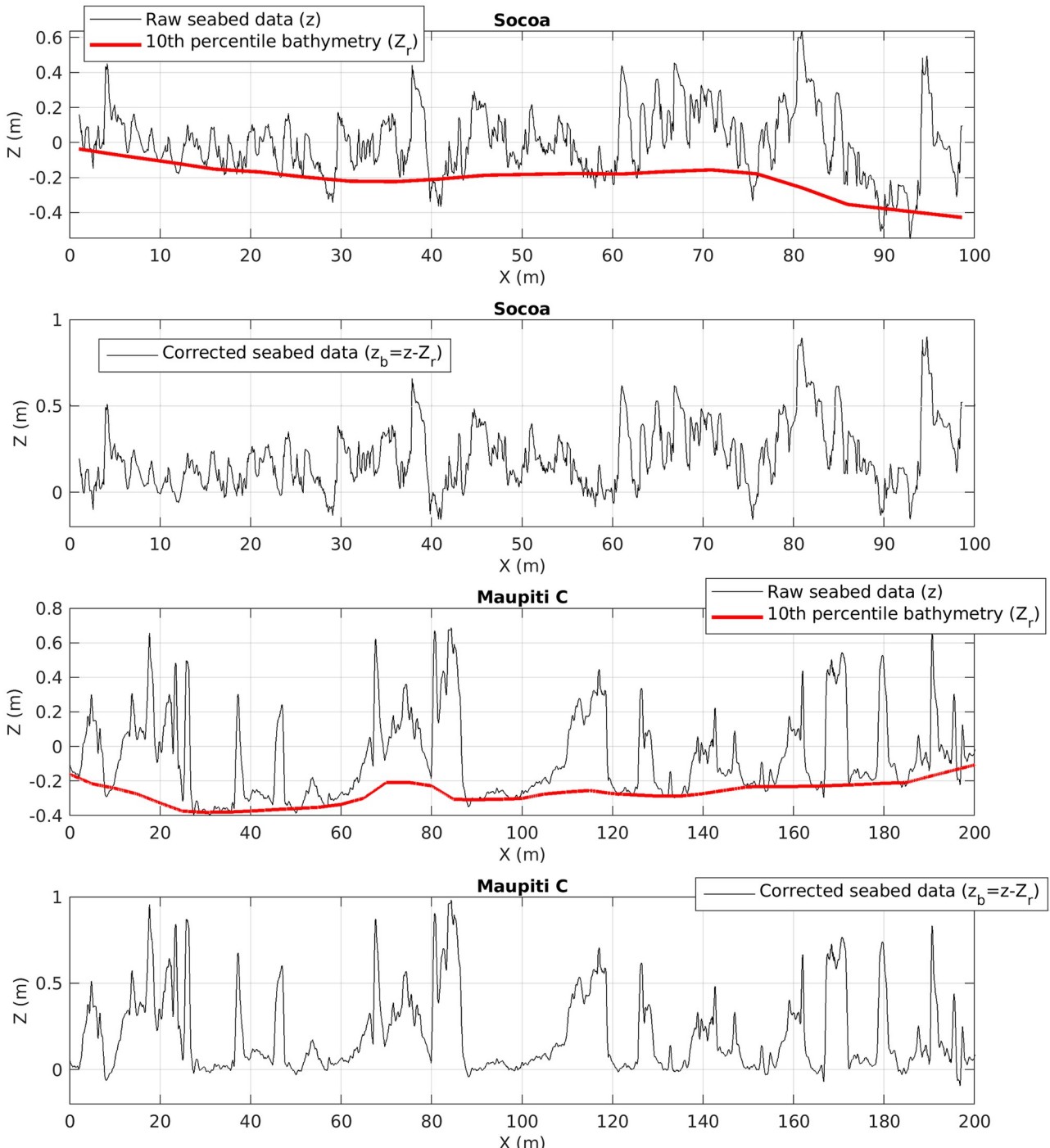

**Fig 7. Illustrative profiles.** Raw seabed data $z$, reconstructed bathymetry $Z_r$ and corrected seabed ($z_b$) for profiles extracted from the Socoa and Maupiti C datasets.

**Skewness.** The 3rd order statistical moment, namely the Skewness, is a measure of the asymmetry of the distribution of seabed elevation:

$$Sk = \frac{\frac{1}{n}\sum_{i=1}^{n}(z_{b,i} - z_m)^3}{\left[\frac{1}{n}\sum_{i=1}^{n}\left(z_{b,i} - z_m\right)^2\right]^{3/2}} \tag{4}$$

Skewness is an estimator for the plan solidity [63]. Skewness is zero for a symmetric distribution. Negative skewness corresponds to a mass of the elevation distribution shifted toward the higher values, i.e. a terrain with closely spaced roughness elements (the so-called *d-type* roughness [68]). Positive skewness corresponds to a mass of the elevation distribution is shifted toward low values, i.e. a terrain with widely spaced roughness elements (the so-called *k-type* roughness).

**Excess kurtosis.** The fourth order statistical moment, namely the Kurtosis, is a measure of the tailedness of the distribution of seabed elevation. We use here the Excess Kurtosis, defining the spread to the normal distribution:

$$EK = \frac{\frac{1}{n}\sum_{i=1}^{n}(z_{b,i} - z_m)^4}{\left[\frac{1}{n}\sum_{i=1}^{n}\left(z_{b,i} - z_m\right)^2\right]^2} - 3 \tag{5}$$

Excess kurtosis is zero for normal distribution. Positive excess kurtosis (leptokurtic distribution) corresponds to a fat-tail distribution, i.e. a larger probability for outliers than the normal-distributed case associated to more developed deep troughs and high peaks in the seabed structure. Negative excess kurtosis (platykurtic distribution) corresponds to compact thin-tail distribution, i.e. lower occurrence of outliers.

**Effective slope.** The *Effective Slope* (*ES*) is defined as the mean absolute gradient in a given direction [64], equivalent to the rate of elevation change [45, 49], also called the neighbor's distance index [6, 65], and to twice the frontal solidity [63]. Over a uniformly spaced terrain profile, it can be written as:

$$ES = \frac{\Delta x}{L}\sum_{i}^{n}\left|\frac{\Delta z_b}{\Delta x}\right|_i \tag{6}$$

where $\Delta x$ is the space step in the *x*-direction.

**Entropy.** *Entropy* is a classical measure of the degree of disorganization of a system, for instance applied to classify oceans floor [23]. The more ordered a signal, the lower the entropy is. For each transect, it is defined as:

$$ENT = -\sum(p_z * log_2(p_z)) \tag{7}$$

where $p_z$ describe the probability distribution.

**Linear rugosity.** The *linear rugosity*, also called rugosity index in ecological studies [66], was calculated as the actual distance accounting for vertical changes, i.e. the sum of the individual distances between successive points, divided by the linear distance between the boundaries

of each selected zone [45, 49]:

$$LR = \frac{\sum_{i=1}^{n-1} \sqrt{(z_{b,i+1} - z_{b,i})^2 + (x_{i+1} - x_i)^2}}{\sqrt{(z_{b,n} - z_{b,1})^2 + (x_n - x_{b,1})^2}} \tag{8}$$

**Coefficient of variation.** The *coefficient of variation* is the mean value of the local ratio between standard deviation and mean values computed between two neighbouring points adapted from [6] (here the absolute value of the mean is used at the denominator):

$$COV = \frac{1}{n-1} \sum_{i=1}^{n-1} \frac{std(z_{b,i:i+1})}{|mean(z_{b,i:i+1})|} \tag{9}$$

**Solid volume fraction.** Related to the plane solidity, the *Solid Fraction* (SF) is the fraction occupied by solid between the maximum and minimum seabed elevation:

$$SF = (z_m - min(z_b))/(max(z_b) - min(z_b)) \tag{10}$$

**Directionality index.** To our knowledge, very few direct estimators of topographical directionality have been proposed in the literature. We use here the directionality index $\Delta$ proposed by [67] based on the computation of directional standard deviations $STD_x$ and $STD_y$, in cross-shore and along-shore directions, respectively:

$$\Delta = \frac{\overline{STD_x} - \overline{STD_y}}{\overline{STD_x} + \overline{STD_y}} \tag{11}$$

## Results

### Statistical analysis

The first part of the analysis is dedicated to the evaluation of seabed elevation spectra, in order to assess to which extent they can be described by the idealized shape from Eq 1. Fig 8 depicts, for each study site, the individual and mean spectra over the selected cross-shore transects. For each considered case, the spectra displays, for part or over the full range of wave numbers, a well-defined power-law structure. The best fit obtained between the observations and an idealized $S \propto K^{\beta}$ spectral model are displayed in red dashed lines. The $\beta$ coefficient ranges from 1.3 to 2.5. At low wave-numbers, a saturation range with a nearly flat spectrum shape is observed for Socoa and the three Maupiti sites. The saturation cut-off is estimated between 0.07 and 0.3 (see solid red lines in Fig 8). For the other sites, the power-law shape extends down to the lowest measured wave-number. Near the upper spectral limit, the spectra displays various behaviors. For Banneg and Socoa, the self-affine range extends up to the high wave-number limit. The three Maupiti spectra show a nearly flat shape at high wave number, maybe related to the accuracy limit of the GNSS method for these sites. A low-pass filter is therefore applied on the Maupiti data in order to remove the high wave-number noise ($k > 4m^{-1}$) for the following statistical analysis. Ars en Ré, Niau, Parlementia A and Parlementia B display a slope change for wave number about $2m^{-1}$, with steeper spectral slopes at higher wave number. This trend may be associated to the progressive transition toward a smooth regime for $K > K_{s2}$ [62].

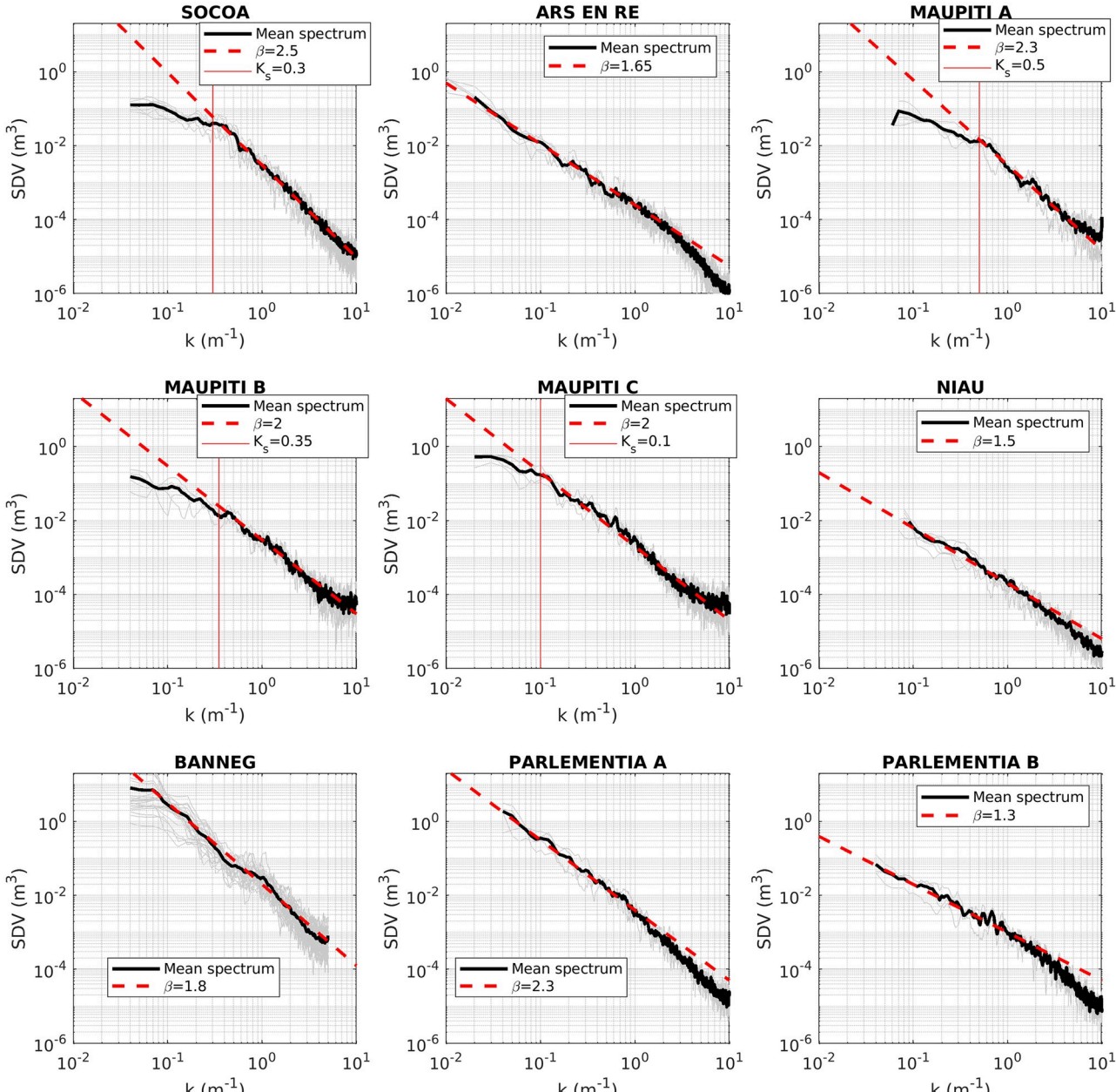

**Fig 8. Compared Spectral Density of Variance (SDV) for the nine study sites.** Grey and black lines indicate the individual and mean spectra, respectively while red dashed and solid lines represent the best-fit power law and the saturation wave number, respectively

Fig 9 depicts the seabed elevation probability density for each study sites. Various histogram shapes are observed from nearly symmetric distributions (Niau) to strong positively skewed structure (Maupiti C, Parlementia A and B, Banneg). This later type of distribution should be associated corresponding to the *k-type* roughness. These observations are quantitatively confirmed by the skewness values obtained in Table 2, reaching 1.42 for Maupiti C. The distribution tailedness also shows variations between sites, in agreement with the excess kurtosis estimations provided in Table 2. Socoa, Maupiti A and Parlementia A and B display a nearly mesokurtic distribution with *EK* values lower than 0.15 (in absolute value). Banneg shows a

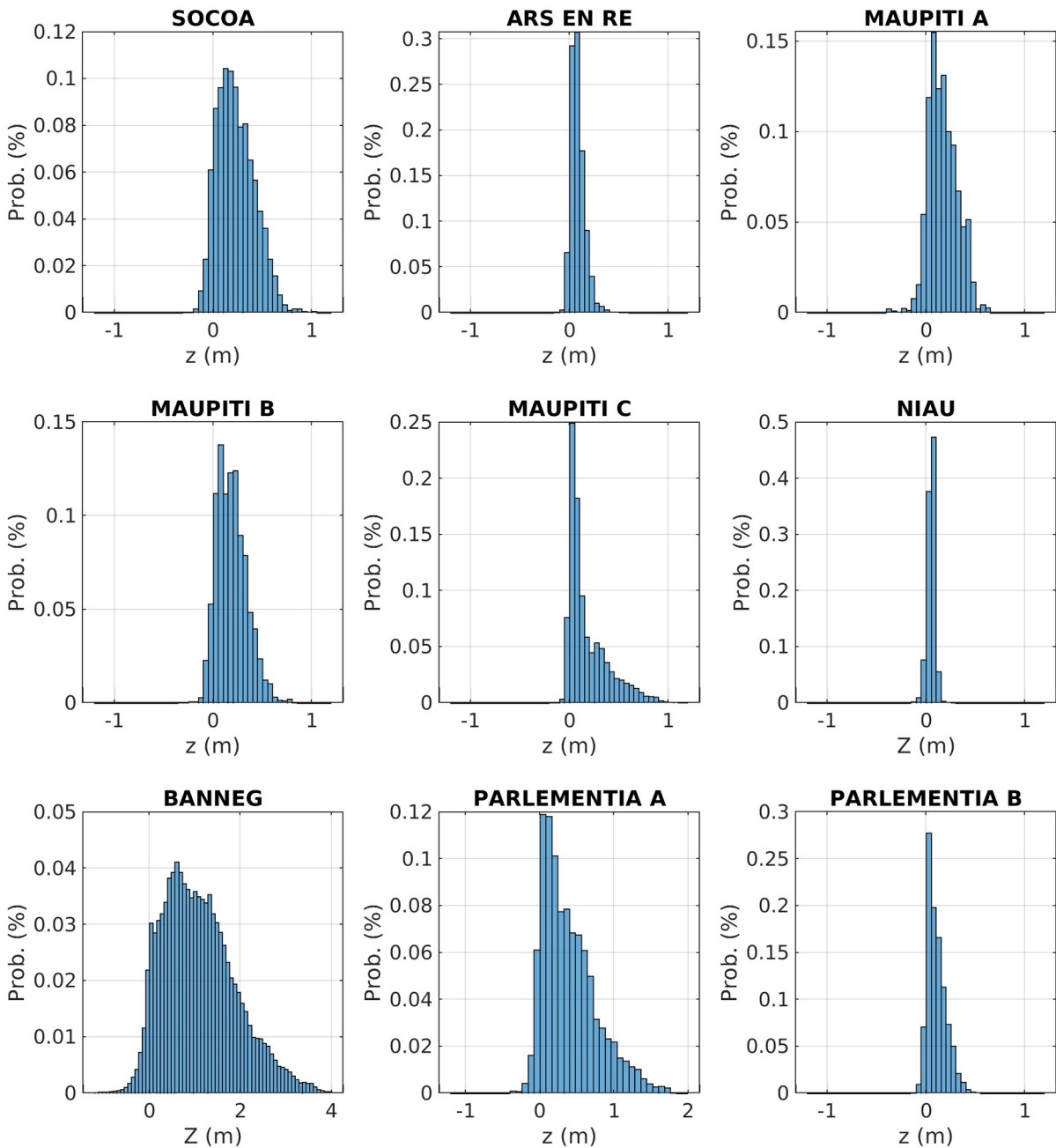

**Fig 9. Compared histograms for the nine study sites.** Note the different x-axis scale for Banneg and Parlementia A.

moderately platykurtic distribution, corresponding to a more squared distribution shape and a reduced occurence of outliers. On the opposite, Niau, Maupiti C and Ars en Ré display a well-developed tailedness with a higher probability of (positive) outliers corresponding to leptokurtic distribution, in agreement with EK values above. Maupiti B shows intermediate level of tailedness, with $EK = 0.47$.

**Table 2. Topographical metrics for the nine study sites.**

| Site | $\beta$ | $K_s$ | Mean | *STD* | SK | EK | ES | ENT | LR | COV | SF | Δ % |
|------|------|------|------|------|------|------|------|------|------|------|------|------|
| Socoa | 1.8 | 0.3 | 0.23 | 0.18 | 0.47 | -0.07 | 0.41 | 3.8 | 1.17 | 0.44 | 0.40 | 25 |
| Ars en Ré | 1.65 | - | 0.079 | 0.064 | 0.88 | 1.65 | 0.13 | 4.57 | 1.02 | 0.15 | 0.38 | - |
| Maupiti A | 2.3 | 0.5 | 0.17 | 0.13 | 0.17 | 0.11 | 0.21 | 3.39 | 1.04 | 0.22 | 0.53 | - |
| Maupiti B | 2 | 0.35 | 0.19 | 0.16 | 0.58 | 0.47 | 0.21 | 3.71 | 1.04 | 0.21 | 0.43 | - |
| Maupiti C | 2 | 0.1 | 0.18 | 0.21 | 1.42 | 1.64 | 0.17 | 3.69 | 1.04 | 0.18 | 0.24 | - |
| Niau | 1.5 | - | 0.05 | 0.04 | -0.47 | 2.13 | 0.14 | 3.62 | 1.02 | 0.19 | 0.55 | - |
| Banneg | 2.2 | - | 1.09 | 0.77 | 0.35 | -0.31 | 0.79 | 3.22 | 1.49 | 0.91 | 0.45 | 2.5 |
| Parlementia A | 2.3 | - | 0.39 | 0.35 | 0.79 | 0.11 | 0.38 | 3.88 | 1.18 | 0.44 | 0.36 | 3 |
| Parlementia B | 1.3 | - | 0.1 | 0.09 | 0.79 | 0.15 | 0.28 | 3.8 | 1.11 | 0.35 | 0.36 | 8 |

## Topographical metrics

The overall aim of the analysis is to identify the optimized set of topographical descriptors able to discriminate between the different sites, i.e. to remove redundant and/or insignificant metrics. The analysis is performed here combining Fig 10 and Table 2, which present the compared metrics for the nine study sites and Fig 11 which graphically depicts the inter-metrics relationships with *r* and *p* indicating the correlation coefficient and the p-value, respectively.

A first overall observation is that each site shows a specific set of values. Interestingly, none of the selected metric is individually able to discriminate between coral and rocky sites. The main observations of each studied metric can be summarized as follows.

- The **spectral slope** $\beta$ shows strong variations ($1.3 < \beta < 2.3$) without straightforward discrimination between coral and rocky sites. The spectral slope $\beta$ does not show clear correlation with other metrics, although a positive correlation with *STD* and Mean may be observed if we exclude Banneg from the regression.

- The **saturation wave-number** $K_s$ shows significant variations between sites, and tends to increase with SF and decrease with SK but overall the $K_s$ dataset is too small to draw statistically robust conclusion.

- The **standard deviation** *STD* displays large variations between sites (normalized standard deviation 102%), which confirms that *STD* constitutes a meaningful indicator of the topographical structure, as assumed by many studies [16, 19, 63]. *STD* is strongly correlated to Mean, owing to the moving window distribution-based filtering of the raw seabed elevation signal. *STD* is also strongly correlated with the linear rugosity LR and the effective slope ES. A statistical connection is also observed with the entropy, but it appears to mostly rely on the large ENT value obtained for Banneg.

- The **mean elevation** *Mean* displays also large variations between sites (normalized standard deviation 116%). It is nearly perfectly correlated with *STD* and shows therefore strong correlation with *STD*-related metrics.

- The **skewness** SK varies between -0.47 and 1.42. Note that the sole negative value, obtained for Niau, is mainly due to the presence of isolated deep crevasses in the reef flat which may impair the discrimination between bathymetry and roughness (see Discussion section). It is disconnected from *STD* but strongly correlated to the solid fraction.

- The **excess kurtosis** EK shows strong variations between sites (normalized standard deviation of 137%) without any straightforward link with other parameters.

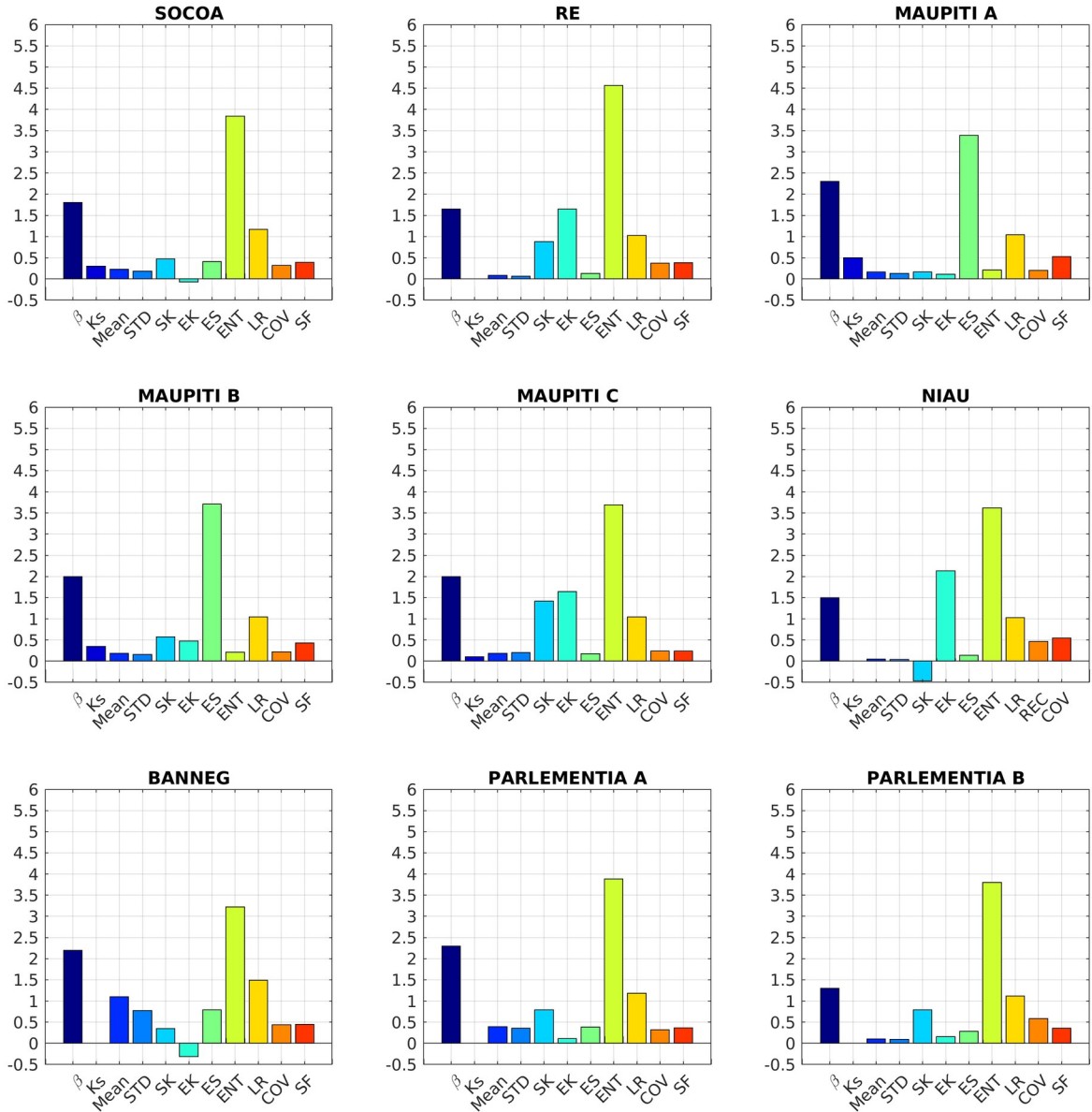

**Fig 10. Compared topographical metrics for the nine study sites.** Note that no $K_s$ is provided for Niau, Ré, Banneg and Parlementia sites, owing to the fact that the self-affine spectral range extends to the lower wave-number limit of the recovered spectra.

- The **effective slope** *ES* varies between 0.13 and 0.79 (normalized standard deviation of 69%). It is well correlated with *STD* and *STD*-related metrics.

- The **entropy** *ENT* displays weak variations between sites with normalized standard deviation of 11%, which decreases to 6% if we discard the Banneg data. *ENT* is statistically connected to *Mean*, *STD* and *LR* but the trends remain fragile and very dependent on the Banneg data.

- The **linear rugosity** *LR* varies from 1.02 to 1.49. It is strongly correlated to *STD*, *Mean* and *ES*.

- The **coefficient of variation** *COV* varies from 0.15 to 0.91. It is only correlated to the spectral slope, with a negative correlation.

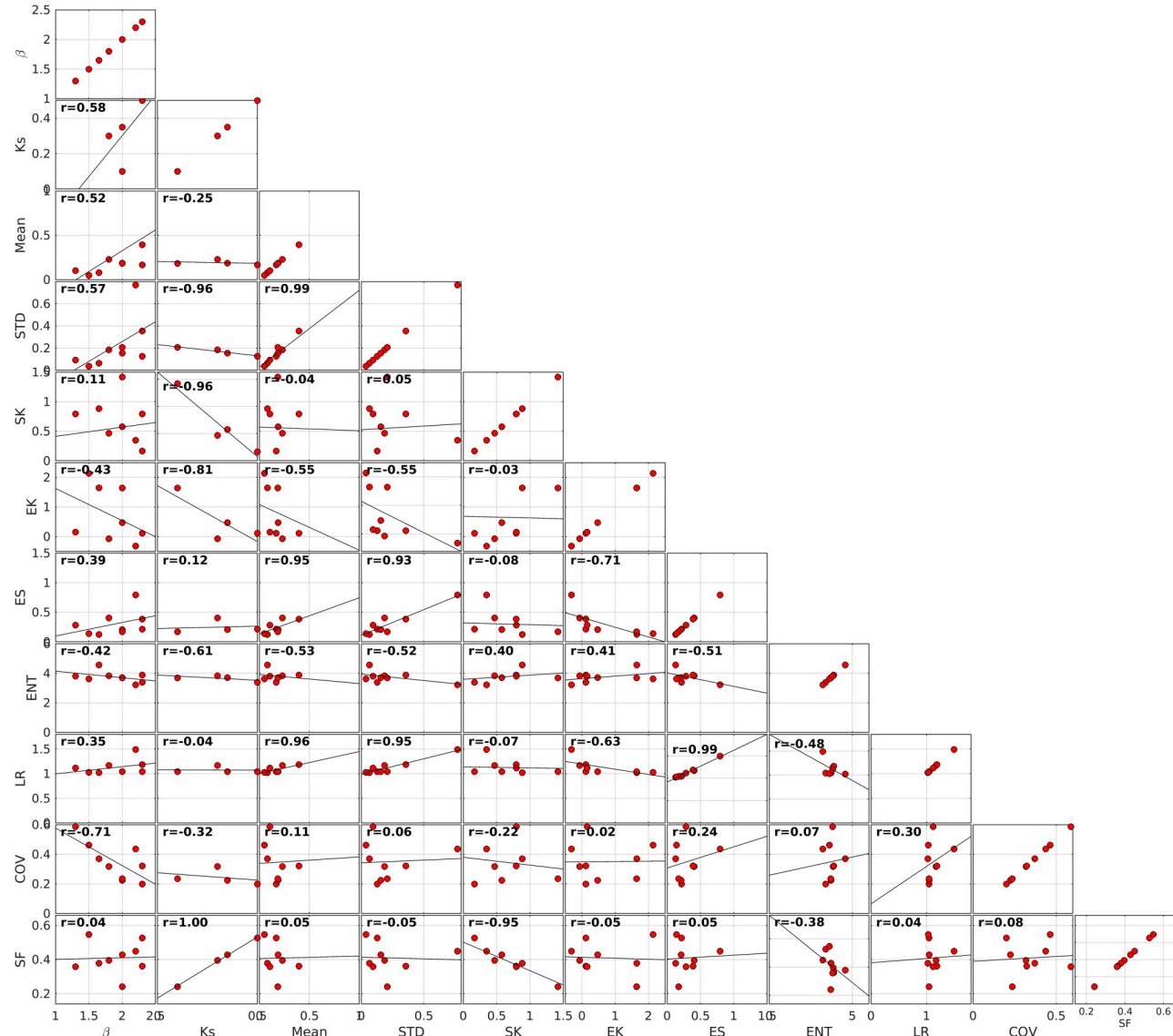

**Fig 11. Inter-metrics relationships.** Field observations are plotted in grey circles. Black solid lines represent the best linear fit. The *r* displays the correlation coefficient.

- The **solid fraction** *SF* varies from 0.24 to 0.55 and is negatively correlated to *SK*

## Directionality

From a qualitative view, the only site showing a clear anisotropy of the topographical structure is Socoa. A subset of four sites, namely Socoa, Banneg, Parlementia A and B have been documented both in cross-shore and alongshore directions (see Table 1) in order to test the ability of the directionality index proposed by [67] to quantify the level of roughness anisotropy.

Computing the directionality index from Eq 11, the values of Δ = 25, 2.5, 3 and 8% for Socoa, Banneg, Parlementia A and B, respectively. The value obtained at Socoa is much larger

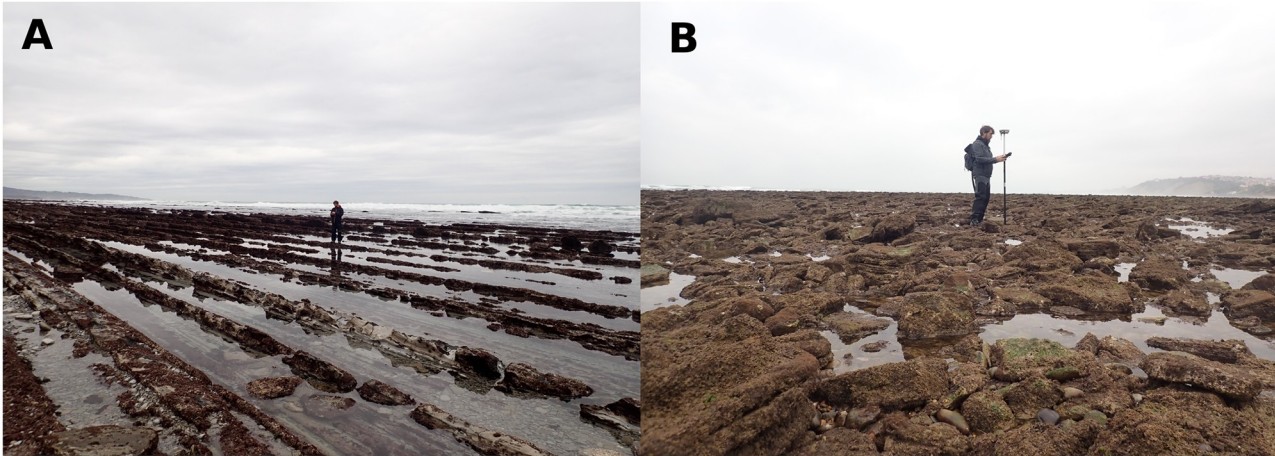

**Fig 12. Illustration of the topography directionality.** Left and right pictures display the typical seabed morphology observed at Socoa and Parlementia A sites, respectively. The corresponding directionality indexes are Δ = 25 and 3%, respectively

than for other sites, which is in good agreement with visual observations (Fig 12). Note that for bidirectional surveys carried out on Socoa and Parlementia A and B sites, the recovered profiles are not perfectly perpendicular, which affects the estimation of the Directionality index. The expected trend is that, for a nearly isotropic topography, the misalignment effect should have negligible consequence on the directionality index computation while, for highly anisotropic sites, the misalignment should lead to underestimate the directionality index. The sensitivity to the profile directions is difficult to quantify from the present set of single profile surveys, but the observed trend is sufficiently strong to differentiate between sites.

## Multi-scale analysis

Fig 13 depicts a multi-scale analysis for each topographical metric, in order to provide further insight on the sensitivity of metrics on the studied domain size. For each studied site, the transects have been subdivided in successive windows on which each metric are locally calculated and then averaged over the whole transect. Three window sizes are tested: 1, 5 and 20m.

The multi-scale analysis reveals a common trend for STD, ENT, ES, LR and COV: a positive correlation with the domain size. This observation should be connected with the fractal spectra observed for all sites in Fig 8. Increasing the extension of the study domain leads to account for larger wave-lengths carrying more topographical variability, which affects each topographical metric related to the magnitude of the topographical fluctuations. It is worthwhile to note that opposite results have been obtained at smaller scale from coral reef data [6], without any straightforward explanation. The distribution-related metrics, i.e. skewness and kurtosis, also show an overall trend to increase at larger scale. This highlights the fact that small scale topographical patterns display a distribution close to the normal one, while the inclusion of larger terrain scales leads to more positively skewed and fat-tailed distribution i.e. more high topographical peaks. SF shows variable response to the analysis scale, either with nearly no sensitivity for Socoa, Ré and Banneg, an increase at larger scale for Niau or a decrease for all other sites. These responses are mostly related to the scaling-induced modification of the statistical distribution of topography which affects the volume calculation: including more high outliers in the data leads to decrease the relative volume occupied by the solid.

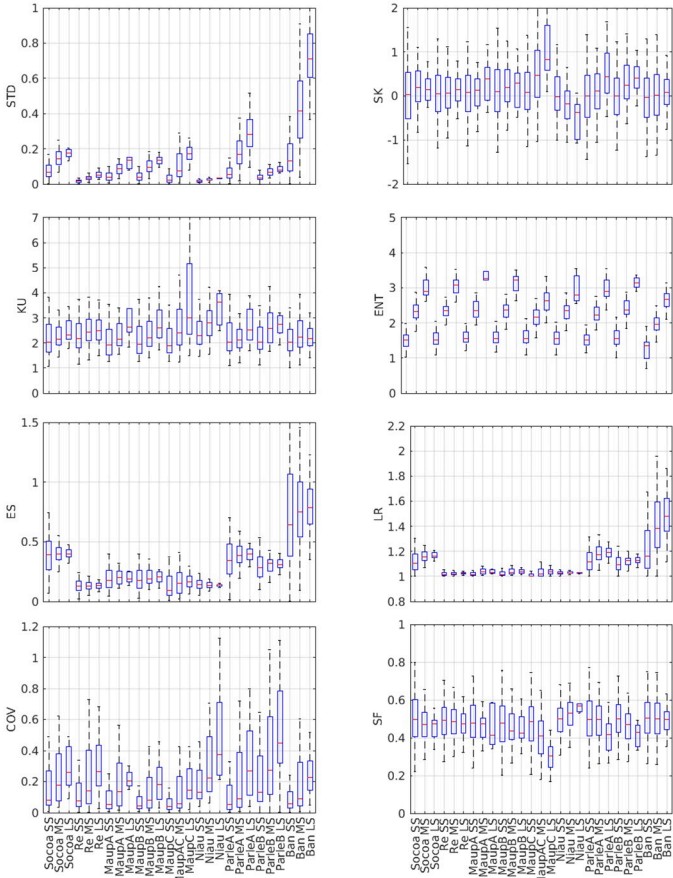

**Fig 13. Multi-scale analysis.** Each plot describes a multi-scale boxplot analysis for a given topographical metric, the horizontal red line describing the median and the y-limits of the box the 25 and 75th percentiles. The x-axis displays the nine studied sites, with three tested scale SS, MS and LS being the small, medium and large scales corresponding to 1, 5 and 20m windows.

## Discussion

The structural complexity of seabed is of primary importance for marine ecosystem health, considering the role played by rocky or coral seabed architecture in forming habitats, and for coastal hazards such as submersion and erosion, in relation to the damping effect of rough seabed on incoming waves. Surveys, management policies and adaptation strategies for coastal zones, including both biodiversity concerns and exposure to physical risks, require to get robust metrics of seabed architecture. Aiming to feed the recent research efforts dedicated to the establishment of quantitative estimators of seabed topographical structure and to provide a more unified view, the present study provides a multi-sites analysis of detailed topographical data over a wide variety of nearshore environments. The study is based on a set of simple topographical metrics, easy to calculate with standard topographical surveys and, for the most part, already used by previous studies in marine ecology or hydrodynamics [6, 23, 49, 62–65]. The compared analysis leads us to recommend a first selection of topographical metrics to be documented in priority. First, the standard deviation constitutes as expected a pivotal metric of the topography heterogeneity, allowing straightforward discrimination between sites. The mean elevation, the linear rugosity and the effective slope show strong statistical connections with

the standard deviation, meaning that they all describe the magnitude of topographical variability. For the sake of standardization and statistical robustness, we may recommend that STD should be preferentially used. Note however that experimental laboratory studies dedicated to bottom drag and wave dissipation [67, 69] on idealized topography have identified a specific, but moderate, effect of effective slope on frictional dissipation, but this would require further confirmation on real terrain. Note also that the relationship between the spectral slope and the effective slope highlighted in the laboratory by [62] is not observed here, probably due to the non-perfect self affine spectra and Gaussian distribution of elevations. Despite its wide use in the ecological field, linear rugosity does not provide fruitful discriminating information at the studied scales. It may still provide useful information at small (sub-metric scale) to characterize habitats [6]. The third and fourth statistical moments, namely the skewness and the excess kurtosis, should deserve attention owing to the fact that they both display also significant differences between sites and, at least for skewness, an effect on friction processes has already been identified [63, 67]. For the studied sites, the solid fraction appears to be strongly correlated to the skewness. Again considering the existing background on skewness [63, 67, 69, 70], we recommend therefore that the former should be discarded in favour of the latter. The spectral parameters (spectral slope, saturation wave-number) show significant differences between sites. It appears necessary to further explore their variation between sites and document their role in friction processes [62]. It is worthwhile to mention that the small-scale (high wave-number) range of the spectra show variable structure, being integrated in the dominant self-affine range for Socoa and Banneg, affected by measurement noise for Maupiti sites or showing a steeper decrease for Ré, Niau and Parlementia A and B sites. This latter decay may be related to a progressive transition toward the expected smooth regime [21, 62]. Entropy and coefficient of variation do not appear to provide meaningful information to differentiate between sites. It should likely be rejected as a direct metric to qualify topographical structure. Finally, the unprecedented analysis of topography structural anisotropy performed here on field data, through the directionality index proposed by [67] from laboratory experiments, show good agreement with visual observations illustrated in Fig 12 and may be recommended for further exploration in other sites.

While time-consuming to build, the present database only gathers nine study sites. Further long-term collective efforts have to be engaged to include more sites in the analysis, in order to enforce the present observations and possibly to extend the range of the selected parameters. A broader database may also lead to identify finer relationships between metrics, for instance through principal component analysis, classification or clustering approaches. In particular, the present analysis has been focused on linear connections between metrics for the sake of interpretability, but non linear relationships may be observed with an extended database. A major challenging issue remains the need to cover wider range of spatial scales than the data presented here and elsewhere in the literature. Many unresolved questions, such as the existence of a smooth regime at small-scale, a saturation regime at large scale or the inconsistency on metric scale-dependency between the present results and previous observations [6], can be only addressed when high-resolution large-extension topographical data will be available, over a variety of comparative sites. As first recommendation, we suggest that four spectral decades should be targeted, with a centimetric resolution over hectometric domains. For now, airborne/underwater 3D laser scanning or photogrammetry are certainly the best approaches to produce such observations [71, 72] and to step beyond the limitations of the present approach, in particular on the directionality issue. The present selection of topographical metrics can therefore be widely applied on the growing amount of high-resolution data, progressing in the long-term development of seabed classification and dynamical monitoring. It is also worth mentioning that the metrics discussed here generally consider the seabed topography as a

surface function $z_b(x, y)$, i.e. the internal volume structure is totally ignored. The extent to which 3D high-resolution survey can reproduce the volumetric structure of seabed topography remains an open question.

Finally, we want to stress that a major obstacle in the unification of the seabed topography studies remains the lack of proper discrimination between bathymetry and roughness. Two particular points deserve further attention and framing in future works. First, the separation between bathymetry and roughness generally relies on a low-pass filter applied on the seabed elevation topography, in order to assign the large/small scale terrain features to the bathymetry/roughness, respectively. The threshold filter length-scale may arise from the properties of fine seabed topography, for example using specific regime transition in the seabed elevation spectra. However, no straightforward length-scale has been observed for the present study sites, the low wave-number saturation region being not present on each site. For coral reef sites, a biological discrimination, with living reef and dead substratum being assigned to roughness and bathymetry, respectively, may deserve further attention but will not be suitable for rocky sites. In a more operational point of view for wave and circulation studies, the filter length-scale between bathymetry and roughness is often dictated by the horizontal resolution of the numerical hydrodynamical model used on the area, with sub-grid features being considered as roughness and their effect on frictional dissipation being parameterized accordingly, while the larger features are explicitly resolved as bathymetry effects. However, this raises two issues. First, the process-based hydrodynamical models generally require several grid points to properly resolve any bathymetric feature. A "grey" zone will therefore exists for terrain features that are too large to be properly described by the subgrid parameterization but still too small to be explicitly resolved by the model. The second main issue of the bathymetry/roughness discrimination based on model resolution will then be that the filter length-scale will be model dependent, i.e. not fixed for a given site. This may not be a problem if addressed with a sound knowledge, but the risk remains that same bathymetry data may be used by different models while it should not be. In the present study, we used an empirical length-scale threshold to discriminate between bathymetry-related and roughness-related terrain features, fixed here at 10 m and kept constant between sites. Additional tests have been carried out with 2, 5 and 20 m filter length. For illustration, Fig 14 displays the dependency of selected roughness metrics on the filter length-scale. The first observation is the strong sensitivity of the four statistical moments on the filter length. As expected, decreasing the filter length leads to a decrease of Mean and *STD*, as part of the seabed variability is transferred from roughness to bathymetry while SK and KU are observed to increase. More stable metric values are obtained when increasing the filter length. This observation may lead to use with large filter length as a more conservative approach. One should bear in mind that it can lead (i) to over-smoothing of bathymetry gradients which are of primary importance for a number of hydrodynamical, geomorphological and ecological processes and (ii) to artifacts in the roughness metrics and/or bathymetry estimation, in particular in the presence of abrupt variations due to crevasses or peaks in the seabed elevation signal.

The second key parameter in the definition of bathymetry is the vertical reference, particularly important for hydrodynamical research works. On one hand, most theoretical frameworks on the frictional dissipation by bottom roughness are based on the definition of a reference base level, namely the bathymetry, *above which* the seabed elevation is defined [63]. On the other hand, most bathymetrical data used in numerical models are reconstructed, as explained above, by low-pass filtering the sounding signal, which results in the production of a reference bathymetry level approximately *in the middle* of the seabed topography. This contradiction is of no consequence as long as the roughness-to-depth ratio remains small. However, for shallow and/or very rough areas, this can lead to significant misestimation of the bottom

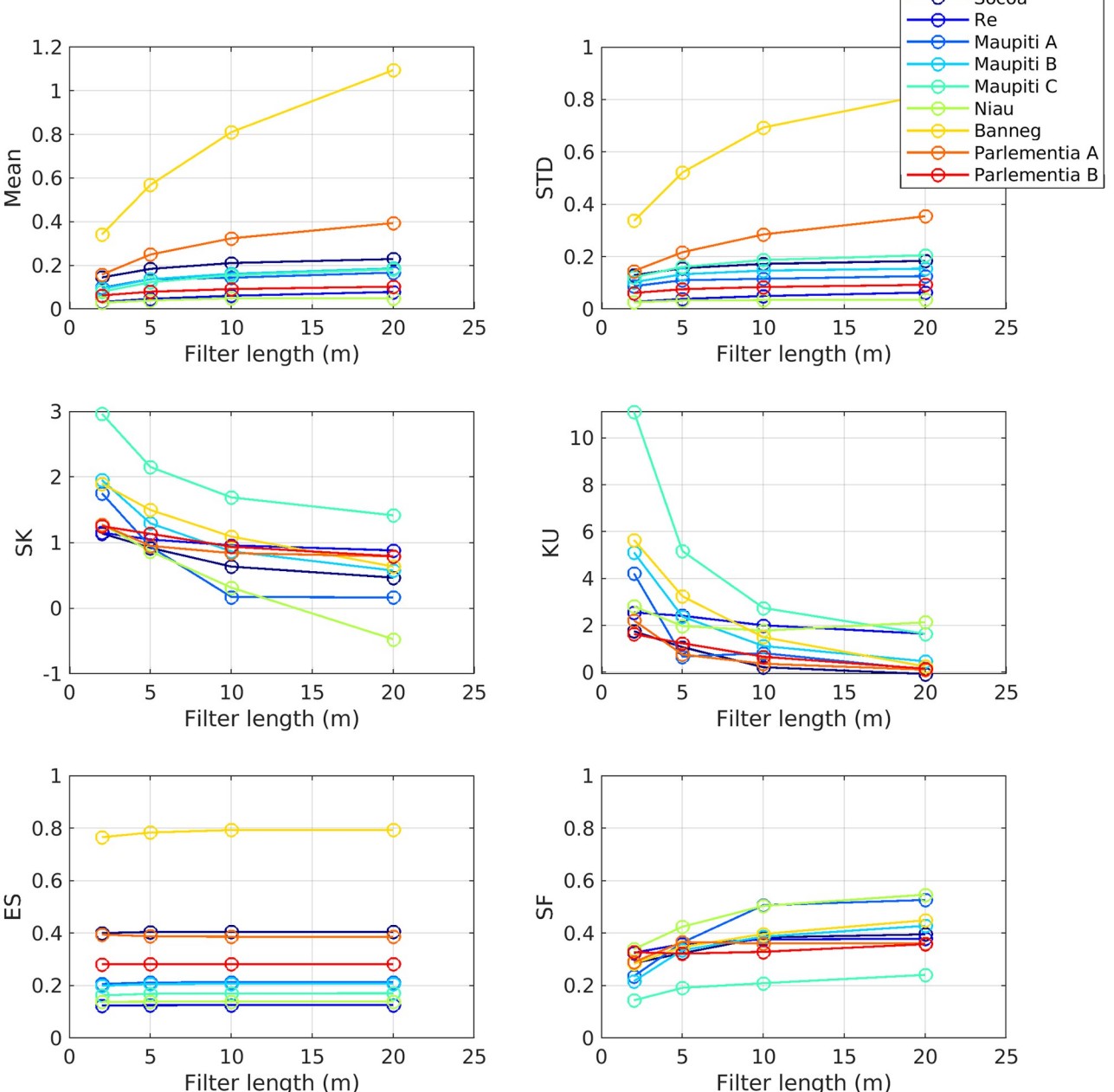

**Fig 14. Metrics sensitivity to the filter length used to discriminate between bathymetry and roughness.**

friction. Simple tests performed on frequency-integrated wave action balance over a 1D 100 m-long horizontal bed, in the sole presence of bottom friction as source term, have demonstrated that wave height are underestimated of 10 to 20% for seabed standard deviation to depth ratio of 0.1 and can be much more at higher roughness height to depth ratio (see Fig 15). The recommendation is therefore made to remain consistent with the base theoretical framework. For areas with documented high-resolution seabed elevation, the bathymetry should be estimated following the present approach, i.e. with a moving-window distribution method

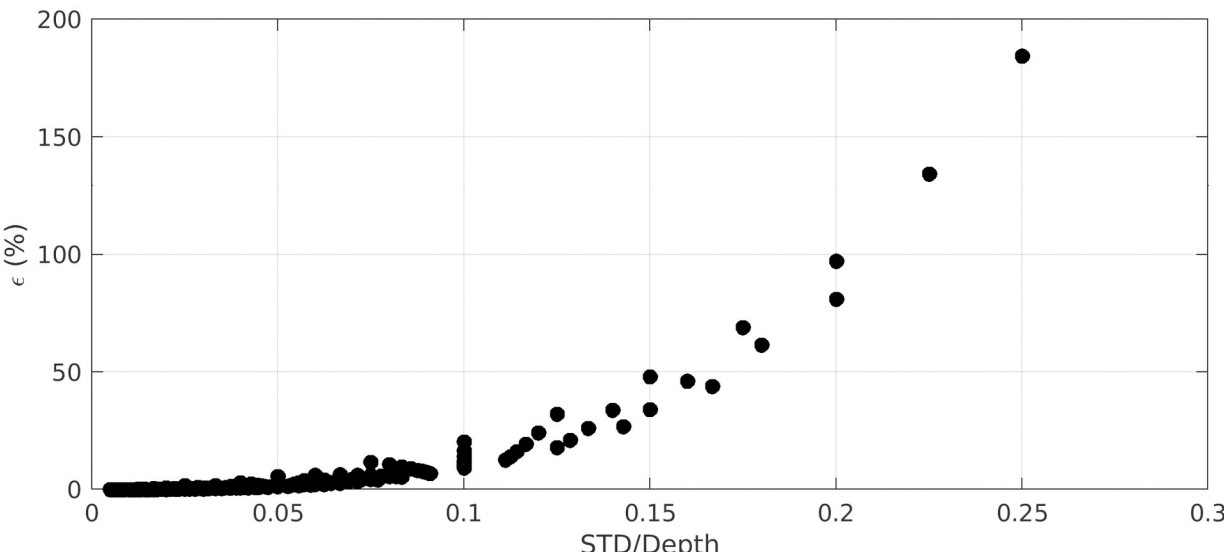

**Fig 15. Effect of bathymetry positioning on wave dissipation versus relative roughness height.** $\epsilon$ is the relative error in significant wave height estimation at the end of the profile when using mean seabed elevation instead of base seabed elevation. STD/Depth is the roughness standard deviation to depth ratio. The estimation is based on a simple 1D frequency-integrated energy balance over a 100 m-long horizontal bottom. Positive error means wave height are underestimated when using the mean seabed elevation compared to the base seabed elevation.

based on a low-valued percentile. Alternatively, when no fine topography data is available, the bathymetry obtained by classical low-pass filtering or moving average should be lowered in rough/shallow areas to reach a relevant base elevation. The lowering magnitude needs *a priori* to be related to the percentile value used to reconstruct the bathymetric: the lower the percentile, the larger the lowering. A maximal value should be around two times the standard deviation, assuming a normal distribution. With the present approach based on the 10-th percentile, the *STD* to Mean relationship plotted in Fig 11 indicates that a bathymetry lowering of 1.2 times the standard deviation of the high-resolution seabed elevation can be used as guideline.

While calling for the acquisition of more fine topographical data, covering both wider spatial scales and site diversity, the present study aims to provide a conceptual and methodological framework to improve our ability to characterize the seabed structure. The establishment of standardized topographical metrics will strengthen the coastal management strategies, both for ecological surveys and restoration plans and for improving the performance of numerical models in predicting the response of coastal systems to submersion events.

## Conclusion

The fine seabed topography receives a growing interest due to its implication in many ecological, hydrodynamical and geomorphological issues. Aiming to lay the basis for a unified framework of the quantitative description of the seabed topographical structure, the present study presents a multi-scale analysis of high-resolution topography surveys in various geomorphological contexts over nine sites, including four coral and five rocky sites. The first main result relies in the selection of relevant and discriminating statistical metrics to describe the topography. The pivotal parameter remains the standard deviation, which allows straightforward differentiation between sites and should be preferred, due to its universal definition, to other metrics describing the magnitude of the topography variability such as linear rugosity or

effective slope. The second parameter of interest is the skewness, which should be preferred over solid fraction. The third metric to be focused on is the spectral shape. The present spectral analysis of topography reveals the ubiquitous presence of a self-affine range in the seabed elevation spectra, with varying extension and slopes between sites. The spectral slopes and regimes clearly deserve extended attention in further studies. Entropy and coefficient of variation does not provided useful information in the present study. A novel metric, named the directionality index, is proposed to quantify the level of topography anisotropy. All selected metrics remain easily calculable and measurable with most survey approaches. Interestingly, none of the selected metrics allowed a direct discrimination between rocky and coral reef sites. A series of recommendations and discussion points are proposed to guide the design of future high-resolution surveys, which are the next necessary step to answer unresolved questions.

## Supporting information

**S1 Appendix. Matlab code.**
(PDF)

## Acknowledgments

The Socoa data has been recovered in the framework of the European FEDER FSE 2014-2020 EZPONDA Program by SIAME and SHOM with GLADYS-UM group support for instruments and human resources. The Ars en Ré data has been retrieved during the RICORE 2020 experiments, supported by SHOM (HOMONIM project and PEA PROTEVS) and the GLADYS-UM group. The Maupiti data has been provided by the MAUPITI HOE 2018 experiments by the GLADYS-UM group. The Niau data has been retrieved with the joint support of the GLADYS group and the Direction de l'Equipement de Polynésie Française. The Banneg data has been extracted from the Litto3D® database, supported by an IGN/SHOM partnership. The Parlementia data has been recovered by the joint efforts of SIAME laboratory and GLADYS group. The Pleiades/SPOT and Sentinel images were acquired as part of the DINAMIS (Dispositif Institutionnel National d'Approvisionnement Mutualisé en Imagerie Satellitaire) project.

The authors are particularly indebted to the colleagues valiantly engaged to recover the on-foot GNSS data: Erik Doerflinger, Benjamin Dubarbier, Marion Pichery, Patrick Marsaleix and Claude Estournel.

## Author Contributions

**Conceptualization:** Damien Sous.

**Data curation:** Damien Sous, Samuel Meulé.

**Formal analysis:** Damien Sous, Ghislain Gassier, Marc Pezerat.

**Funding acquisition:** Damien Sous.

**Investigation:** Samuel Meulé, Héloïse Michaud.

**Methodology:** Damien Sous, Samuel Meulé, Solène Dealbera, Héloïse Michaud, Ghislain Gassier, Marc Pezerat, Frédéric Bouchette.

**Project administration:** Damien Sous, Héloïse Michaud, Frédéric Bouchette.

**Resources:** Samuel Meulé, Frédéric Bouchette.

**Supervision:** Damien Sous, Frédéric Bouchette.

**Validation:** Damien Sous, Samuel Meulé, Solène Dealbera, Héloïse Michaud, Marc Pezerat, Frédéric Bouchette.

**Writing – original draft:** Damien Sous.

**Writing – review & editing:** Samuel Meulé, Solène Dealbera, Héloïse Michaud, Ghislain Gassier, Marc Pezerat, Frédéric Bouchette.

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
