## [Decision Letter · Decision Letter 0]

1 Feb 2024

PONE-D-23-30056Quantifying the topographical structure of rough seabedsPLOS ONE

Dear Dr. Sous,

Thank you for submitting your manuscript to PLOS ONE. After careful consideration, we feel that it has merit but does not fully meet PLOS ONE’s publication criteria as it currently stands. Therefore, we invite you to submit a revised version of the manuscript that addresses the points raised during the review process.

We look forward to receiving your revised manuscript.

Kind regards,

Md. Naimur Rahman

Academic Editor

PLOS ONE

Journal Requirements:

4. Please note that funding information should not appear in any section or other areas of your manuscript. We will only publish funding information present in the Funding Statement section of the online submission form. Please remove any funding-related text from the manuscript.

5. We note that you have indicated that there are restrictions to data sharing for this study. PLOS only allows data to be available upon request if there are legal or ethical restrictions on sharing data publicly. For more information on unacceptable data access restrictions, please see http://journals.plos.org/plosone/s/data-availability#loc-unacceptable-data-access-restrictions. 

6. We note that Figures 1-6 in your submission contain map/satellite images which may be copyrighted. All PLOS content is published under the Creative Commons Attribution License (CC BY 4.0), which means that the manuscript, images, and Supporting Information files will be freely available online, and any third party is permitted to access, download, copy, distribute, and use these materials in any way, even commercially, with proper attribution. For these reasons, we cannot publish previously copyrighted maps or satellite images created using proprietary data, such as Google software (Google Maps, Street View, and Earth). For more information, see our copyright guidelines: http://journals.plos.org/plosone/s/licenses-and-copyright.

a. You may seek permission from the original copyright holder of Figures 1-6 to publish the content specifically under the CC BY 4.0 license.  

Additional Editor Comments:

Dear Authors,

Thank you for submitting your manuscript to our journal. We have completed the review process, and I have carefully considered the feedback provided by the reviewers and my own reading of your paper.

Your study presents a comprehensive multi-site analysis of high-resolution topography surveys in rough nearshore environments, with the objective of identifying relevant metrics for quantifying the geometrical structure of seabeds. The use of advanced ANN models combined with remote sensing data in your methodology is commendable and represents a significant step forward in geomorphological research.

However, there are several areas that require attention before your manuscript can be considered for publication:

1. **Introduction and Contextualization**: The introduction needs to be expanded to better situate the study within the existing body of literature. It should clearly articulate the research gaps your study aims to address. Please ensure that both historical and recent relevant studies are cited to demonstrate the evolution of the field and to justify the need for your research.

2. **Methodological Details**: While your methodology is robust, greater detail is required to ensure reproducibility. Specifics on data collection, processing, and analysis should be expanded. Please consider including supplementary material if necessary to accommodate additional methodological explanation.

3. **Statistical Analysis**: The statistical approach used to identify relevant roughness metrics needs a more detailed explanation. The choice of metrics and their computation warrants further elaboration. Furthermore, a discussion on the implications of choosing different metrics would be beneficial to the reader.

4. **Results Presentation**: The results section would benefit from additional figures and tables that synthesize the key findings. Visual representations of the data can greatly enhance the readability and impact of your results.

5. **Discussion**: The discussion could be strengthened by directly linking the study's findings to the implications for regional planning and mitigation strategies. Additionally, the potential limitations of the study and directions for future research should be more thoroughly discussed.

6. **Conclusion**: The conclusion should succinctly summarize the main findings, their significance, and the contribution to the field. It should also reflect on the broader implications of the work.

7. **References**: Ensure that all references are up-to-date and relevant. Any old references should be replaced with the latest research where applicable.

8. **Data Availability**: The manuscript must adhere to our data availability policies. Please provide a clear statement regarding where the data supporting your findings can be accessed.

9. **Graphics Quality**: Some of the figures provided are not of high enough quality for publication. Please ensure that all figures are clear, high-resolution, and correctly annotated.

10. **Reviewers' Comments**: Please address all specific points raised by the reviewers in your revision. A detailed point-by-point response will be required to explain how you have addressed each comment or why a suggestion was not incorporated.

In conclusion, your manuscript has the potential to make a valuable contribution to our understanding of seabed topography. I invite you to revise your manuscript in line with the feedback provided and resubmit for further consideration. We look forward to receiving your revised manuscript.

Sincerely,

Md. Naimur Rahman

Reviewers' comments:

Reviewer's Responses to Questions

**Comments to the Author**

1. Is the manuscript technically sound, and do the data support the conclusions?

Reviewer #1: Yes

2. Has the statistical analysis been performed appropriately and rigorously? 

Reviewer #1: No

3. Have the authors made all data underlying the findings in their manuscript fully available?

Reviewer #1: Yes

4. Is the manuscript presented in an intelligible fashion and written in standard English?

Reviewer #1: Yes

5. Review Comments to the Author

Reviewer #1: =================

General Comments:

The present work aims to report on a comparative multi-site analysis of high-resolution topography surveys in rough nearshore environments, particularly those with rocky and coral seabeds. This analysis contributes to proposing a reduced set of relevant roughness estimators, with the aim of defining a unified framework for reconstructing roughness statistics and bathymetry from fine seabed topographical data.

For this purpose, the authors studied nine sites and quantified the geometrical structure of seabeds using various metrics. With this information, they attempted to identify representative estimators capable of quantitatively discriminating between seabed types.

The paper is well-organized, but some results need improvement, and certain issues require clarification. Some figures are of good quality, but others need improvement, as explained below in the specific comments.

Despite the need for some clarifications in the reviewer's opinion, this article should be interesting to the readership of PLOS ONE. The reviewer recommends the article for publication, with consideration given to the specific comments listed below.

Specific Comments:

The title appears too general, referring only to rough seabeds. I suggest the title should clarify whether the focus is specifically on rocky and coral seabeds.

L.180: The authors state, "The final selected profile 180 lengths are given in Table 1. T." In which column? This information is not provided there.

L. 299-300: The last sentence of the paragraph is poorly written. Please improve this.

L. 307-312: This analysis is poorly presented with incorrect values. The values mentioned in the paragraph do not correspond to those in Table 2, and it is unclear, based on the positive or negative values of kurtosis, why, for example, Maupiti A is considered a platykurtic distribution and Banneg a leptokurtic distribution. Please, carefully check this.

Figs. 10 and 11: The resolution of the PDF figures is poor, improving when opening the images in JPG. However, even with improved quality, the font size of letters and numbers is too small for correct reading. Please, improve it. Additionally, the caption of Figure 10 states, "The r displays the determination coefficient," but despite being difficult to discern, I believe what is being presented is the correlation coefficient.

L. 337-370: The values of the directionality index from Equation 11 are presented. However, when examining the respective initial figures (e.g., Fig.5), cross-shore and alongshore profiles appear to have been conducted with some obliquity to the coastline. The authors should discuss this because they make no mention of it and should also address the implications of this obliquity for the results.

L. 405: "they all describe" not "they all describes."

L. 434: The authors state, "the directionality index proposed by [51] from laboratory experiments, show good agreement with visual observations," but I believe no visual observations are presented before. Are these visualizations observable, i.e., can they be measured from images that corroborate the values of Equation 11? This should be clarified.

L. 525: Conclusion: The conclusions are presented in a very general manner. It is necessary to specify the conclusions reached by this work in relation to the objectives they set out to achieve. Please, rewrite this section more clearly.

6. PLOS authors have the option to publish the peer review history of their article (what does this mean?). If published, this will include your full peer review and any attached files.

Reviewer #1: No

---

## [Decision Letter · Decision Letter 1]

3 Apr 2024

PONE-D-23-30056R1Quantifying the topographical structure of rocky and coral seabedsPLOS ONE

Dear Dr. Sous,

Thank you for submitting your manuscript to PLOS ONE. After careful consideration, we feel that it has merit but does not fully meet PLOS ONE’s publication criteria as it currently stands. Therefore, we invite you to submit a revised version of the manuscript that addresses the points raised during the review process.

We look forward to receiving your revised manuscript.

Kind regards,

Md. Naimur Rahman

Academic Editor

PLOS ONE

Journal Requirements:

**Additional Editor Comments:**

Please address all the comments from the reviewer carefully.

Reviewers' comments:

Reviewer's Responses to Questions

**Comments to the Author**

1. If the authors have adequately addressed your comments raised in a previous round of review and you feel that this manuscript is now acceptable for publication, you may indicate that here to bypass the “Comments to the Author” section, enter your conflict of interest statement in the “Confidential to Editor” section, and submit your "Accept" recommendation.

Reviewer #1: (No Response)

2. Is the manuscript technically sound, and do the data support the conclusions?

Reviewer #1: Yes

3. Has the statistical analysis been performed appropriately and rigorously? 

Reviewer #1: Yes

4. Have the authors made all data underlying the findings in their manuscript fully available?

Reviewer #1: Yes

5. Is the manuscript presented in an intelligible fashion and written in standard English?

Reviewer #1: Yes

6. Review Comments to the Author

Reviewer #1: The answers and revisions to the paper seem to have been satisfactorily addressed. However, this verification cannot be viewed entirely because the revised paper has a problem with the figures.

In particular:

- The authors mention that they have introduced two additional new figures, totaling 15 figures. However, only 13 figures appear, as in the previous version of the paper;

- The figures appear in a disordered manner, and some are not always numbered, making direct analysis with the text difficult (13, 12, 9, 8, 7, 6, 5, 2, 1, ?, ?, 4 and 3!);

- Even the ones that are ordered do not correspond to the text's description. For example, at the beginning of the Results section, it mentions figure 8, but in the document, this seems to correspond to the figure indicated as number 7.

Therefore, the reviewer recommends correcting what has been described so that the article can be considered for publication.

7. PLOS authors have the option to publish the peer review history of their article (what does this mean?). If published, this will include your full peer review and any attached files.

Reviewer #1: No

---

## [Author Response · Author response to Decision Letter 1]

4 Apr 2024

We are sorry for the inconvenience caused by the incorrect figure location which has been corrected in the new version of the paper. 

Best regards

---

## [Decision Letter · Decision Letter 2]

25 Apr 2024

Quantifying the topographical structure of rocky and coral seabeds

PONE-D-23-30056R2

Dear Dr. Sous,

We’re pleased to inform you that your manuscript has been judged scientifically suitable for publication and will be formally accepted for publication once it meets all outstanding technical requirements.

Kind regards,

Md. Naimur Rahman

Academic Editor

PLOS ONE

Additional Editor Comments (optional):

Reviewers' comments:

Reviewer's Responses to Questions

**Comments to the Author**

1. If the authors have adequately addressed your comments raised in a previous round of review and you feel that this manuscript is now acceptable for publication, you may indicate that here to bypass the “Comments to the Author” section, enter your conflict of interest statement in the “Confidential to Editor” section, and submit your "Accept" recommendation.

Reviewer #1: All comments have been addressed

2. Is the manuscript technically sound, and do the data support the conclusions?

Reviewer #1: Yes

3. Has the statistical analysis been performed appropriately and rigorously? 

Reviewer #1: Yes

4. Have the authors made all data underlying the findings in their manuscript fully available?

Reviewer #1: Yes

5. Is the manuscript presented in an intelligible fashion and written in standard English?

Reviewer #1: Yes

6. Review Comments to the Author

Reviewer #1: The answers and revisions to the paper are satisfactorily addressed and I consider that the paper is acceptable for publication.

7. PLOS authors have the option to publish the peer review history of their article (what does this mean?). If published, this will include your full peer review and any attached files.

Reviewer #1: No

---

## [Editor Report · Acceptance letter]

1 May 2024

PONE-D-23-30056R2 

PLOS ONE

Dear Dr. Sous, 

I'm pleased to inform you that your manuscript has been deemed suitable for publication in PLOS ONE. Congratulations! Your manuscript is now being handed over to our production team.

Kind regards, 

on behalf of

Mr Md. Naimur Rahman 

Academic Editor

PLOS ONE